# CO Electroreduction Mechanism on Single-Atom Zn (101) Surfaces: Pathway to C2 Products

**DOI:** 10.3390/molecules28124606

**Published:** 2023-06-07

**Authors:** Yixin Wang, Ming Zheng, Xin Zhou, Qingjiang Pan, Mingxia Li

**Affiliations:** 1MIIT Key Laboratory of Critical Materials Technology for New Energy Conversion and Storage, School of Chemistry and Chemical Engineering, Harbin Institute of Technology, Harbin 150001, China; 21s025052@stu.hit.edu.cn (Y.W.); 19b925114@stu.hit.edu.cn (M.Z.); 2Key Laboratory of Functional Inorganic Material Chemistry, Ministry of Education of the People’s Republic of China, School of Chemistry and Materials Science, Heilongjiang University, Harbin 150080, China; panqingjiang@hlju.edu.cn

**Keywords:** density functional theory, electrocatalysis, mechanisms, CO_2_RR, CO ethane

## Abstract

Electrocatalytic reduction of carbon dioxide (CO_2_RR) employs electricity to store renewable energy in the form of reduction products. The activity and selectivity of the reaction depend on the inherent properties of electrode materials. Single-atom alloys (SAAs) exhibit high atomic utilization efficiency and unique catalytic activity, making them promising alternatives to precious metal catalysts. In this study, density functional theory (DFT) was employed to predict stability and high catalytic activity of Cu/Zn (101) and Pd/Zn (101) catalysts in the electrochemical environment at the single-atom reaction site. The mechanism of C2 products (glyoxal, acetaldehyde, ethylene, and ethane) produced by electrochemical reduction on the surface was elucidated. The C-C coupling process occurs through the CO dimerization mechanism, and the formation of the *CHOCO intermediate proves beneficial, as it inhibits both HER and CO protonation. Furthermore, the synergistic effect between single atoms and Zn results in a distinct adsorption behavior of intermediates compared to traditional metals, giving SAAs unique selectivity towards the C2 mechanism. At lower voltages, the Zn (101) single-atom alloy demonstrates the most advantageous performance in generating ethane on the surface, while acetaldehyde and ethylene exhibit significant certain potential. These findings establish a theoretical foundation for the design of more efficient and selective carbon dioxide catalysts.

## 1. Introduction

Unlike limited fossil fuels, which are produced from solar energy stored underground in ancient times, renewable electricity can come from a number of environmentally friendly sources, such as wind, hydro, and solar. Replenishing fuels only reduces the carbon-containing material. If the released CO_2_ can be chemically reduced to value-added products, and electricity can provide the power for the reaction, then any clean energy can be stored in fossil fuels, simultaneously addressing issues for intermittent power supply from these sources [1]. Since the observation of HCOOH in the cathode Hg in 1870, many electrocatalysts have been screened for their effectiveness in this reaction [2,3,4,5,6,7]. Carbon capture, utilization, and storage (CCUS) technology is one of the most promising methods to effectively reduce carbon emissions. As a potential hydrocarbon chemical, the electrocatalytic CO_2_ reduction reaction (CO_2_RR) can be accomplished via different mechanisms, and produces various products, including C1 products (e.g., CO, formic acid, methane, methanol) and C2+ products (e.g., ethylene, ethanol, acetaldehyde, acetic acid, glyoxal, propylene, n-propyl alcohol, etc.) [8,9]. So far, the generation of C1-C2 products has achieved excellent results with high faradaic efficiency (FE) [6,10,11,12]. However, due to the complex mechanism of the proton–electron transfer process, the selectivity towards the formation of C2 products is still highly challenging [13]. The species depend on the electrocatalytic material [14], and the electrocatalytic design needs to overcome several key bottlenecks, including high overpotential and low selectivity of target products [15].

Hori et al. [16] first reported that Cu electrodes can reduce CO_2_ into C1 products and C2 products. Studies on the generation of multi-carbons have attracted attention. A number of CO_2_RR electrocatalysts have been reported, including mono-metal, bimetallic, oxides, and some two-dimensional materials with unique properties [17,18,19,20,21,22]. Zinc is abundant in the earth’s crust and has high selectivity for CO products. Zn has the potential to reduce CO_2_ to CO, and is considered a substitute for Au, Ag, and other precious metals. The pure metal block of Zn has low electrocatalytic CO_2_ reduction activity, so the FE of Zn has been improved by adjusting size and surface [23,24]. The major crystal surface for forming CO products is the zinc catalyst (101), which is different from H_2_ generation on (002). The linear relationship between Zn (101) and (002) crystal face ratios with CO FE can be adjusted to obtain different products, such as syngas with a controllable molar ratio (CO/H_2_: 0.2–2.31) [25,26] on the surface of Zn, and the FE can reach 85%. In particular, the more open Zn (101) surface exhibits greater selectivity to CO than the Zn (002) facet. For CO_2_RR hydrocarbon products, obtaining high value-added products using zinc remains an issue. Zn (101) can convert CO_2_ into hydrocarbons by a one-pot tandem reaction with other elements [27,28,29]. Another metal captures the CO generated on the surface of Zn and produces value-added products through the CO reduction reaction (CORR). Recently, the rapid development of single-atom alloys (SAAs), which have unique properties, has facilitated the exploration of CO_2_RR mechanisms and the design of novel catalysts [30]. In addition, SAAs have garnered significant attention in the scientific community due to their exceptional catalytic activity, which surpasses that of traditional catalysts [31]. By modifying the neighboring single-atom sites, SAAs has the ability to enhance the selectivity of CO_2_RR [32,33]. These ideas have sparked interest in identifying possible mechanisms for CO_2_RR and the basic steps that control product distribution.

C–C coupling is a key step in the reduction of CO to C2 products. Several mechanisms have been proposed to elucidate the specific conversion pathways of CO to ethylene, ethanol, acetaldehyde, acetic acid, and other polycarbon products. Calle-Vallejo and Koper proposed that CO dimerization is the initial step in the formation of C2 products [34]. Expanding on this hypothesis, Cheng et al. established a detailed coupling mechanism, where, after CO dimerization, oxygen atoms undergo hydrogenation (*OCCO + H → *COHCO) [35]. Subsequently, Goodpaster et al., employing the CEP model, observed that *CHO and *CO dimerize to form *CHOCO [36]. This reaction takes place following the reduction of *CO to *CHO at higher potential (e.g., U ≈ −1 V). Utilizing DFT calculations, Bell et al. provided an explanation for the mechanisms of seven C2 products [37]. In thermodynamic terms, the outcome of the competition for electrocatalytic reactions is determined by the stability of the first hydrogenation intermediates (*OCCOH or *OCCHO for C2 production, *H for HER) [38,39]. For example, if *CHOCO or *COHCO is more stable than *H, it will occupy the active sites, and the primary pathway will be the generation of C2 products, whereas *H is more stable favoring HER. As a result, the key to enhancing C2 production and suppressing H_2_ production is to discover electrocatalysts that bind *CHOCO or *COHCO more strongly than *H (ΔG_*H_ > ΔG_*OCCOH_ or ΔG_*OCCHO_). Bell et al. [38] proposed a one-pot tandem reaction for CO_2_RR, where CO_2_ is first reduced to CO by gold (or silver), and then CO is captured and further reduced to C1 by a single atom. There is an opportunity to explore electrocatalysts that can first convert CO_2_ to CO, and then to hydrocarbons, while suppressing HER. As previous studies have shown that CO_2_ is first converted into CO before C2 generation, and the electroreduction results of CO are similar to those of C2 products from CO_2_, we have chosen to focus on the mechanism of CO as a starting point [40,41]. While the mechanism of CO_2_ to CO on Zn materials has been extensively studied, the mechanism of CO to C2 compounds still requires further exploration. Our research aims to investigate this process, as it has been shown to determine the selectivity of CO_2_RR [36].

The purpose of this study is to investigate the electrochemical reduction mechanism of CO on Pd/Zn (101) and Cu/Zn (101) by analyzing the potential products of glyoxal, acetaldehyde, ethylene, and ethane. Our goal is to gain a deeper understanding of the thermodynamic mechanisms of producing C2 products on SAAs. In this way, we aim to develop SAAs that are highly selective for C2 products, leading to more efficient and sustainable processes for the production of valuable chemicals. This study provides direction for the implementation of novel SAA materials and holds immense importance in addressing the global environmental and energy dilemma.

## 2. Results and Discussion

### 2.1. Material Selectivity and Stability

The electrochemical stability of a catalyst plays a crucial role in determining its catalytic efficiency. In this study, we have conducted calculations and provided a detailed description of the electrochemical stability of M/Zn catalysts [42]. In ref. [42], it was found that out of the 16 SAAs studied containing Fe, Ru, Os, Co, Rh, Ir, Ni, Pd, Pt, Cu, Ag, Au, V, Mo, Ti, and Cr, only these doped with Rh, Ir, Pd, Pt, Cu, Ag, and Au show stability. Further analysis of the ΔG_*CO_ and E_ads*CO_ revealed that Cu/Zn (101) and Pd/Zn (101) exhibit a high electrochemical reduction selectivity for CO, positioning them as the most stable and selective catalysts for further research. Based on these findings, we selected Cu/Zn (101) and Pd/Zn (101) as catalysts to investigate the reduction of CO to C2 products, and explore the corresponding reaction mechanism on their respective surfaces.

In addition, recent theoretical studies have revealed that individual metal atoms can be covered by *H, *OH, or *O, where the symbol * represents the surface site, depending on the pH and electrode potential [43]. Considering the strong binding ability of *OH to metal atoms and the pathway of *O protonation leading to the formation of *OH intermediates during the CO_2_ reduction process, we analyzed the influence of *OH on the stability of surface-active sites. If *OH adsorption is too strong, it can passivate the reaction site, rendering it inactive. Therefore, we calculated the lowest surface potential for the M/Zn + H_2_O → OH-M/Zn + H^+^ + e^−^ (M = Cu, Pd) reaction at different pH values, as depicted in Figure 1. When the pH and potential fall within the yellow region, M/Zn (101) can maintain its monatomic active site. Conversely, in the gray area, the single-atomic active site is passivated by *OH, making it difficult to adsorb CO molecules. It is evident that as the reaction environment becomes more alkaline, a higher potential is required to sustain the active site of the monatomic reaction. Currently, the desired CO_2_ reduction potential is 0.70 eV, allowing for a broad range of pH values to be selected, which can provide reaction sites for single atoms. Moreover, the activities of Pd/Zn (101) and Cu/Zn (101) can be guaranteed at pH = 0. Moreover, Pd/Zn (101) (pH = 0, U < 0.50 V) has a wider voltage range compared to Cu/Zn (101) (pH = 0, U < 0.22 V). the introduction of Pd single atoms has a more favorable effect on the stability of Zn materials than the introduction of Cu single atoms. However, the addition of single Cu atoms to Zn materials has a more significant impact on improving their ability to selectively adsorb CO molecules compared to the addition of Pd atoms to Zn materials. This improvement is reflected in the decrease in the ΔG for CO adsorption, which is 0.32 eV for Cu/Zn (101) and 0.35 eV for Pd/Zn (101). Therefore, we will investigate the reduction mechanism of these two materials for C2 production.

We present the stable region of potential and pH value, which enables the determination of the pH range based on the CO_2_ reduction potential on different catalysts, as well as the prediction of the reaction solution type. For example, Appendix A shows the minimum voltage required by the seven stable M/Zn (101) catalysts to sustain the monatomic active sites under different pH values of common solutions. Moreover, by analyzing the surface state, we pre-evaluate the active site stability of monatomic materials in the experimental environment.

To produce C2 products, CO plays a critical role as an important intermediate. Previous studies have highlighted the significant role of Zn site in reducing CO_2_ to CO through a one-pot series reaction over M/Zn (101) [25]. Initially, CO_2_ molecules adsorb onto the catalyst’s surface, and subsequently, the adsorbed CO_2_ receives two pairs of electrons and protons resulting in the formation of *CO. Finally, carbon monoxide and water are released from the surface. The free energies of Zn (101) and Zn (002) during the reduction of CO_2_ are shown in Figure 2. The first protonation step (*CO_2_ → *COOH) serves as the potential determining step (PDS) with respective ΔG values of 0.70 eV and 1.22 eV. Our calculated results agree well with the experimental results, demonstrating that the reaction proceeds as follows: CO_2_ → *CO_2_ → *COOH → *CO + H_2_O [25], and the ΔG value of PDS is similar to the calculated value of this experimental result.

Assuming CO as the reaction precursor, we propose that CORR proceeds at the atomic site. The results of our calculations support the assumption that the E_ads_ of CO_2_ on the Zn surface is −0.05 eV, while at the Cu single-atom site it is 0.08 eV, and at Pd it is −0.02 eV. This suggests that the binding effect between CO_2_ and Zn is stronger than that of single-atom sites. Furthermore, the ΔG of CO_2_ on the Zn surface is 0.23 eV, while on Pd sites it is 0.39 eV, and on Cu it is 0.40 eV. These findings indicate that the Zn site is more conducive to adsorbing CO_2_ molecules, which in turn leads to reduction reactions. Specifically, E_ads*CO_ on Zn is −0.23 eV, with ΔG = 0.25 eV. E_ads*CO_ on Pd is −0.58 eV, with ΔG = −0.09 eV. Finally, E_ads*CO_ CO on Cu is −0.51 eV, with a ΔG of −0.03 eV. Therefore, single-atom sites exhibit remarkable selectivity for *CO adsorption. We explore the reduction of CO to C2 products at the single-atom sites of Pd/Zn (101) and Cu/Zn (101). By investigating the reaction mechanisms and optimizing the reaction conditions, we hope to shed light on the fundamental principles underlying this important process.

### 2.2. C–C Coupling Mechanism

We studied the mechanism of C–C bond formation, specifically focusing on three intermediate species: *CHO, *COH, and *CO. Mechanism I involves the dimerization of *CO, resulting in the formation of the *OCCO intermediate [34]; Mechanism II entails the coupling of protonated *CO intermediates (*CHO and *COH) to generate *CHOCO or *COHCO [37]; and Mechanism III involves a coupling reaction between intermediates (*C and *CH_2_) formed through the multi-step protonation of *CO [44]. Other reaction intermediates, such as *CHOCHO, *(CH_2_)_2_, *COHCOH, *COHCHO, and others, may also contribute to C–C coupling. However, due to their low surface concentration, these mechanisms are unlikely to have a significant impact on the process. To determine the mechanism of C–C bond formation and the species involved in this process, we calculated the ΔG for different coupling and protonation steps on Cu/Zn (101) and Pd/Zn (101) surfaces, as illustrated in Figure 3.

Mechanism I: Through calculations, it was found that the formation of OCCO through *CO coupling only overcomes a ΔG of 0.43 eV (Pd/Zn) and 0.17 eV (Cu/Zn), which corresponds to a lower free energy compared to the coupling of *CHO/*COH intermediates for C–C bond formation. As a result, it exhibits a thermodynamic advantage.Mechanism II: Analysis of the ΔG for various protonation and coupling processes of intermediates on the catalyst surface reveals that while *COH and *CHO are favorable for coupling with CO to form C–C bonds, the reduction of *CO to *CHO or *COH at lower voltages is challenging. Consequently, the generation of C–C bonds through Mechanism II does not offer an advantage.Mechanism III: Since the formation of *CHO requires a high ΔG, the mechanism involving CH_2_ dimerization to form C2 products is not advantageous. Even at high potentials, the presence of an *CHO intermediate is possible, but the ΔG for further hydrogenation of the oxygen atom is significant, at 0.48 eV (Pd/Zn) and 0.70 eV (Cu/Zn), rendering the involvement of CH_2_ in C–C coupling unrealistic. The protonation of the oxygen atom in *COH to generate C also encounters obstacles; therefore, Mechanism III is not considered on the catalyst surface.

*CO dimerization is more likely to occur at higher CO densities, [45] and * OCCO is observed [46]. After calculating the specific ΔG of the CO dimerization process, the results are depicted in Appendix A. The calculations indicate that the initial potential for *OCCO formation is −0.43/−0.17 eV (vs. HER). The mechanism of *CO dimerization can occur on Cu/Zn (101) and Pd/Zn (101) surfaces at relatively low potentials. In the *CO → *OCCO → *CHOCO pathway, the formation of the C–C bond represents the potential-determining step [36], which does not involve protons and aligns with the observed pH dependence experimentally [47].

The electronic structure directly affects the stability of intermediates and indirectly determines the surface reduction mechanism. To explore the charge behavior between the substrate and *OCCO, we calculated the differential charge density and Bader charge of *OCCO adsorbed on the surface, and the results are illustrated in Figure 4. The formation of the *OCCO intermediate occurs at the surface, where the CO moiety of *OCCO near the surface is bound to a Pd atom and three Zn atoms. The carbon–oxygen bond length is elongated compared to that of carbon monoxide, indicating activation of *OCCO on the catalyst surface. Analysis of differential charge density diagram reveals that the surface donates electrons to the *OCCO intermediate. The contribution of specific orbitals can be seen in Appendix A. The adsorption of *OCCO by Cu/Zn (101) and Pd/Zn (101) is comparable, with the only distinguishing factor being the difference in DOS broadening of the d orbitals due to different types of single atoms. However, this difference does not affect the adsorption of *OCCO. M/Zn (101) exhibits orbital interaction with *OCCO, and provides electrons for the π* orbitals of *OCCO, enabling stable adsorption and activation of *OCCO. Bader results indicate that there is a total charge transfer of 1.027 and 0.9614, respectively, for the four atoms directly interacting with *OCCO, and the adsorption state of the dimer has more electrons than the gas state. Consequently, the dimer carries a negative charge in the adsorption state, which becomes more stable after solvation, with an energy difference of around 0.03 eV. This adsorption configuration resembles the CO dimer structure observed on the Cu (100) surface by the Koper group [34], except in this case, the interacting surface is an alloy material, or even a non-Cu system. This suggests that Zn single-atom alloys exhibit similar chemical behavior to Cu, which can be explained by their comparable electronic structures [48]. Furthermore, it implies that Cu materials, which possess the highest potential for CO_2_ reduction, can potentially be substituted with Zn single-atom alloys, and these alloys may demonstrate unique selectivity for CO_2_ reduction products.

### 2.3. The Competition of *OCCO Reduction on Cu/Zn (101) and Pd/Zn (101)

We analyzed two protonation modes of OCCO to determine their selectivity; protonation of the oxygen atom generates the *COHCO intermediate, while protonation of the carbon atom generates the *CHOCO intermediate. Our calculations on the Zn single-atom alloy surface show that the *CHOCO intermediate is energetically favored due to its lower potential energy, as shown in Figure 3. The energy difference between *CHOCO and *COHCO on the Zn single-atom alloy surface is 0.67 eV (Pd/Zn) and 0.86 eV (Cu/Zn), which is consistent with Bell’s results. However, the energy difference is larger than the result on the Cu surface (0.43 eV) [37]. This is likely due to the interaction between oxygen atoms and Zn on the surface.

To investigate the competition between the C–C coupling reaction and the HER, we calculated the ΔG* values for the intermediates *CHOCO, *OCCHO, and *H. The results showed that the hydrogenation process is more favorable after the formation of C–C bonds, indicated by negative ΔG* values (−0.13 eV for Pd/Zn and −0.26 eV for Cu/Zn). Previous work has calculated ΔG_*H_ > 0 [25], thus ΔG_*OCCHO_ < ΔG_*H_. This suggests that the catalyst exhibits higher selectivity for *OCCO hydrogenation, which inhibits the HER to some extent. We also found that the C1 and C2 mechanisms compete for the common intermediate *CHO, which is involved in both C2 products and methane generation. The result of ΔG_*OCCO_ < ΔG_*CHO/ΔG*COH_ indicate that at low potential, the production of C1 products through *CHO is less thermodynamically favorable than through *CHOCO, resulting in a higher production efficiency of C2 products. However, the efficiency of producing C1 products through *CHO increases with increasing potential. Our calculations indicate that the competition between the C2 and C1 mechanisms on the surface of Zn single-atom alloys is similar to that on Cu (100). Experimental and theoretical evidence suggests that C2 products are favored on Cu (100) at low voltage. Zn single-atom alloys may utilize the *CO dimerization mechanism to generate C2 products, and the synergistic effect between single atoms and Zn atoms may make the performance of Zn single-atom alloys superior to that of Cu catalysts.

The key step in the C–C coupling process on the surface of M/Zn (M = Pd, Zn) catalysts via the *CO dimerization mechanism is the protonation of the carbon atom, leading to the formation of the *CHOCO intermediate, which exhibits high selectivity. Experimental evidence supports the existence of the *CHOCO intermediate [49]. In comparison to the protonation of the C1 product in the initial step at a lower potential, the formation of the C2 product is more prominent, while HER is suppressed.

### 2.4. The Pathway of Generating C2 Products

It has been established that CO_2_ is converted to CO before undergoing further reduction, and the electroreduction products of CO are the same as those of CO_2_ [36,40,41,50,51,52,53]. Furthermore, while the mechanism of CO_2_ to CO conversion has been extensively studied, the mechanism of CO to C2 compounds is still insufficient. Therefore, our study focuses on the mechanism starting from CO rather than CO_2_. The reduction of CO results in the production of four C2 products, namely glyoxal, acetaldehyde, ethylene, and ethane, when using M/Zn (101) (M = Cu, Pd) as a catalyst. Selective analysis was conducted for each product, and the intermediate structures are presented in Figure 5. Among them, ethane (represented by the green pathway) exhibits the highest thermodynamic advantage, while glyoxal (purple pathway) has the lowest. Acetaldehyde (chartreuse pathway) and ethylene (orange pathway) are identified as secondary products. By using Cu/Zn (101) as an example, we can analyze the specific mechanism of intermediate selection, which is also applicable to Pd/Zn (101).

After coupling with CO on the surface, the carbon atoms outside the surface undergo hydrogenation to form a *CHOCO intermediate. The C–C bond length of *CHOCO is 1.401 Å, which falls between the C–C bond and C=C bond lengths. *CHOCO stretches the C–O bonds to 1.372 Å and 1.371 Å, respectively, approaching the carbon oxygen single bond length, indicating activation of the intermediate. At lower voltages, the dominant pathway involves the formation of *CHOCO and hydroxyl groups, with the *CHCO retaining a double-bond length of 1.33 Å. The next two steps involve protonation on two carbon atoms, alternately generating * OCHCH_2_. The proton–electron transfer in this step is crucial, as it determines the reaction mechanism of ethylene and ethane.

#### 2.4.1. Ethane

The most favorable thermodynamic pathway is the ethane mechanism. In this mechanism, the methylene carbon atom of *OCHCH_2_ is protonated, leaving *O to be hydrogenated into water. There are two possible cases of protonation in the intermediate *OCH_2_CH_3_: (i) the oxygen atom is protonated to obtain the ethanol group (ΔG = 0.25 eV), or (ii) the carbon atom is protonated to form ethane, which then desorbs. In the first case, for the desorption of OCH_2_CH_3_ on the Cu/Zn (101) surface and reduction of ethanol to its desorbed state, a minimum voltage of −0.84 eV is required. This suggests that the ethanol produced during the green mechanism process in Figure 5 will remain on the surface. This assumption is supported by the calculated adsorption energy, which shows that ethanol binds to the surface through oxygen atoms, causing the O–C bond to elongate from 1.440 Å to 1.453 Å, with E_ads_ = −0.43 eV. Therefore, ethanol is further adsorbed and reduced to ethane. In the second case, the reduction of CO_2_ to ethane involves a transfer of 14 electrons, while the process of generating ethanol involves a transfer of 12 electrons. Based on the length of the reaction chain, it appears that the process of generating ethane is more susceptible to side reactions than the ethanol process. The free energy for generating ethane is ΔG = −0.38 eV, which is 1.22 eV lower than the free energy for generating ethanol. By comparing the free energy, it can be concluded that the second protonation to ethane is a thermodynamically advantageous process, with a higher selectivity for ethane. This thermodynamic advantage can offset some of the side reaction effects caused by the lengthening of the reaction chain. During the process of ethane generation, the hydrogenation of the methyl group in *OCH_2_CH_3_ results in the production of methane, which is a C1 product. The internal energy barrier for this process is 0.15 eV. Additionally, the path from 7 to 8 of the C2 product generates a ΔG of −0.67 eV, making it difficult for the C–C bond to break and produce methane. As a result, *OCH_2_CH_3_ continues to reduce and generate C2 ethane with high selectivity.

#### 2.4.2. Ethylene

The ethylene path is an alternative reaction mechanism (shown as the orange path in Figure 5). There are two ethylene paths: (i) protonation of the oxygen atom of OCHCH_2_, which generates ethylene desorption, or (ii) protonation of the methylene carbon atom, which causes cleavage of the C–O bond to generate ethylene. However, on the surface, these steps are hindered by the presence of double bonds, which are typically energetically unstable and therefore unfavorable for the reaction to occur. In the first case of oxygen atom hydrogenation (Figure 6: 6 → 13, 0.26 eV), the thermodynamics favor the cleavage of the C–O bond to generate the *CHCH_2_ pathway. This preference may be attributed to the adsorption ability of Zn on hydroxyl groups and the synergistic effect of Zn and M on *CHCH_2_. In the second case of carbon atom hydrogenation (Figure 6: 6 → 6b, 0.55 eV), the Cu surface generally favors the cleavage of the C–O bond on *OCH_2_CH_2_, resulting in abundant ethylene on its surface [34]. However, the synergistic effect of Zn and Cu atoms causes *OCH_2_CH_2_ to bind to the surface through both carbon and oxygen atoms, making it difficult for ethylene to desorb. The production of ethylene is an important step in the synthesis of many industrial chemicals. The Cu/Zn (101) surface has been found to have a catalytic effect on this process, with the protonation of the oxygen atom of * OCHCH_2_ playing a key role (as shown in Figure 6 on the orange path). As shown in Appendix A, the situation is the same for Pd/Zn (101).

In the process of ethane and ethylene production, there exist three crucial intermediates that have a significant impact on the selectivity of the C2 products. Firstly, intermediate 6 can produce either ethylene or ethane depending on whether protonation occurs on the O or C atom. Although ethane has a thermodynamic advantage, the difference in free energy (ΔG) between the two reactions is only 0.21 eV (Cu/Zn) and 0.29 eV (Pd/Zn). Since the ethane pathway requires two additional electron transfer steps compared to the ethylene pathway, the efficiency of the two reactions may be similar. Secondly, intermediate 8 can be hydrogenated on either the C or O atom to produce ethane or ethanol, respectively. The ethanol pathway requires overcoming a ΔG of 0.25 eV (Cu/Zn) and 0.19 eV (Pd/Zn) to generate desorbed ethanol, while the ethane pathway has a ΔG of −0.38 eV (Cu/Zn) and −0.37 eV (Pd/Zn), indicating a significant advantage in producing ethane. Thirdly, intermediate 7 can produce either methane or ethane depending on whether methyl protonation occurs. The energy difference between the two pathways is 0.82 eV (Cu/Zn) and 0.88 eV (Pd/Zn), indicating a high selectivity for producing ethane.

#### 2.4.3. Glyoxal and Acetaldehyde

Currently, the amount of glyoxal present in the C2 product generated by CO_2_RR is minimal [40]. Figure 5 reveals that intermediate 10 of * CHOCO reduction is easily consumed because the next step of generating acetaldehyde (step 10 → 11 → 12) after the protonation of the carbon atom is thermodynamically favorable. The reduction of CHOCO and the desorption of acetaldehyde on the Cu/Zn (101) surface to vacuum glyoxal (steps 3 → 13) require at least a high voltage of −0.90 eV, indicating that most of the glyoxal does not desorb during the reaction process. The calculated adsorption energy supports this assumption: trans glyoxal is more stable in the gas phase, with an energy difference of 0.20 eV compared to cis glyoxal. The experimental and calculated results are consistent [54]. Cis glyoxal is stably bound to the surface by two sp^2^ hybrid oxygen atoms, forming a C=C bond between the carbon atoms, with an adsorption energy of −1.64 eV. This electron rearrangement reaction shortens the C–C bond length of glyoxal from 1.547 Å in the gas phase to 1.382 Å in surface adsorption. Therefore, surface adsorption of glyoxal belongs to chemical adsorption, which adsorbs on the surface for the next step of hydrogenation reduction. When glyoxal is reduced, it produces an intermediate known as an acetaldehyde group (shown as intermediate “11” in Figure 5). To convert this intermediate into acetaldehyde, only a small amount of energy (ΔG = 0.14 eV) is required. Furthermore, the desorption of acetaldehyde occurs spontaneously (ΔG = −0.65 eV), indicating that the surface is conducive to diffusion once acetaldehyde is formed. However, acetaldehyde cannot act as an intermediate in the generation of ethylene, as evidenced by the fact that ethylene was not detected in the glyoxal reduction experiment [55].

Although the mechanisms on the surfaces of Cu/Zn (101) and Pd/Zn (101) are similar, there is a notable difference in the selectivity of Cu/Zn (101) and Pd/Zn (101) to ethane due to the unique properties of surface ΔG. Specifically, the higher selectivity of Cu/Zn (101) to ethane can be attributed to the distinct characteristics of surface G. The specific ΔG value is shown in Appendix A.

## 3. Computational Methods

We used the Vienna ab initio Simulation Package (VASP) to perform our calculations [56,57]. The Perdew-Burke-Ernzerhof (PBE) exchange-correlation function and projector-augmented wave pseudo-potentials were employed in the calculations [58]. The plane-wave basis cutoff energy was set to 450 eV, and we sampled the Brillouin zone using a 3 × 3 × 1 k-points mesh according to the Gamma scheme for the slab calculations. We used a convergence criterion of 10^−5^ eV for electronic relaxation and fully relaxed the structures until all residual forces on each atom were less than 0.03 eV·Å^−1^. The calculations took into account spin-polarization and z-direction dipole correction.

As shown in Appendix A, the slab models used to simulate the catalytic reaction consisted of five metal layers, with each layer containing a (4 × 4) periodic cell. The bottom three layers remained fixed during the process. To prevent interactions between periodic replicas along the *z*-axis, a 15 Å vacuum separation was used between adjacent images. The geometries at different steps were optimized.

The adsorption energy, E_ads_, defined as E_ads_ = E_TM-slab_ − E_TM_ − E_slab_, is calculated, where E_TM-slab_, E_TM_, and E_slab_ represent the total energy of the adsorbate-slab (TM-slab), a gas molecule TM, and the catalyst (slab), respectively. The Gibbs free energy (ΔG) of each elementary step was calculated using the computational hydrogen electrode (CHE) model, which was developed by Nørskov et al. [59]. The model is defined by the equation ΔG = ΔE_DFT_ + ΔE_ZPE_ + TΔS + ΔG_u_ + ΔG_pH_, where ΔE_DFT_ represents the reaction energy, which can be directly obtained from the total energy of DFT. ΔE_ZPE_ and ΔS are the free energy correction and entropy, respectively. T represents the system temperature (298.15 K), while ΔG_u_ is the contribution of the applied electrode potential pair ΔG. ΔG_pH_ = k_B_T × pH × ln10, which is only dependent on changes in H concentration. The pH level does not affect the PDS of the catalytic reaction. To facilitate the comparison of the catalytic performance of SAAs, we set ΔG_pH_ = 0. We used the implicit model in VASPsol to deal with solvation effects [60,61], and the dielectric constant of water used in the calculation was 78.4.

## 4. Conclusions

In this study, we analyzed the stability and internal energy barriers of different reaction intermediates in the bifurcation process of CO electrochemical reduction to C2 products on Zn (101) single-atom alloys using DFT calculations and literature-reported data. Our results demonstrate that the C–C coupling process on the surfaces of Cu/Zn (101) and Pd/Zn (101) can be achieved using the CO dimerization mechanism at lower voltages. Protonation to generate *CHOCO after C–C coupling exhibits higher selectivity than HER and *CO protonation. The surfaces of Cu/Zn (101) and Pd/Zn (101) play a crucial role in stabilizing the reaction by transferring electrons to the intermediate. Introducing single atoms compensated for the disadvantage of Zn materials, enabling them to adsorb and reduce CO. Both Cu/Zn (101) and Pd/Zn (101) surfaces effectively reduce CO to ethane, acetaldehyde, and ethylene at lower voltages. The interaction between the single atoms and Zn on the surface of Cu/Zn (101) and Pd/Zn (101) creates a unique catalytic environment that enhances selectivity for C2 intermediate products, such as ethane. This work utilizes the synergistic effect of SAAs in the mechanism of generating C2 products. Therefore, the CO_2_ reduction products generated have high selectivity. This provides inspiration for SAAs in different catalytic fields, such as starting from the target reaction and artificially selecting the expected single atom to be doped with the substrate. This study contributes to the understanding of the CO_2_ reduction mechanism on Zn metal catalysts and the design and development of highly selective SAAs catalysts.

## Figures and Tables

**Figure 1 molecules-28-04606-f001:**
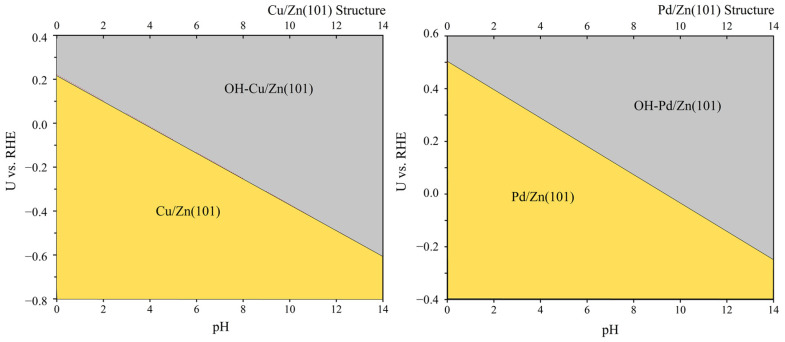
The potential required for Cu/Zn (101) and Pd/Zn (101) to remain stable at different pH values in the reaction of M/Zn + H_2_O → OH–M/Zn + H^+^ + e^−^.

**Figure 2 molecules-28-04606-f002:**
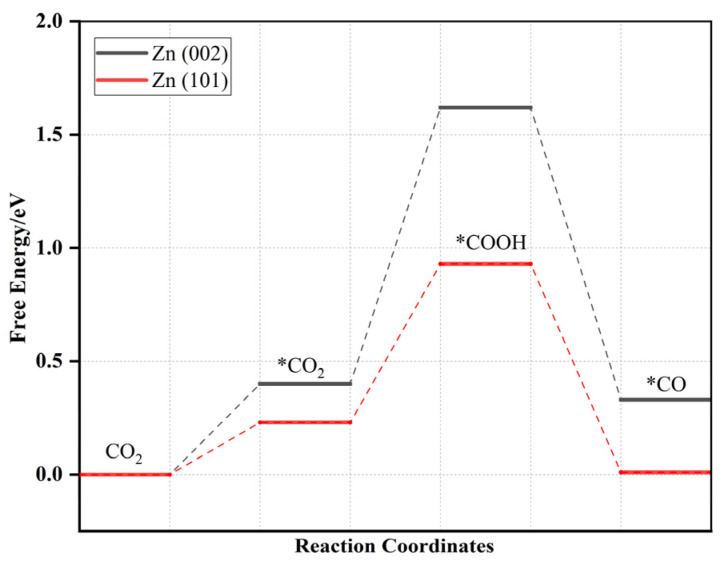
Free energy diagram for CO_2_ to CO on Zn (002) (black line) and Zn (101) (red line).

**Figure 3 molecules-28-04606-f003:**
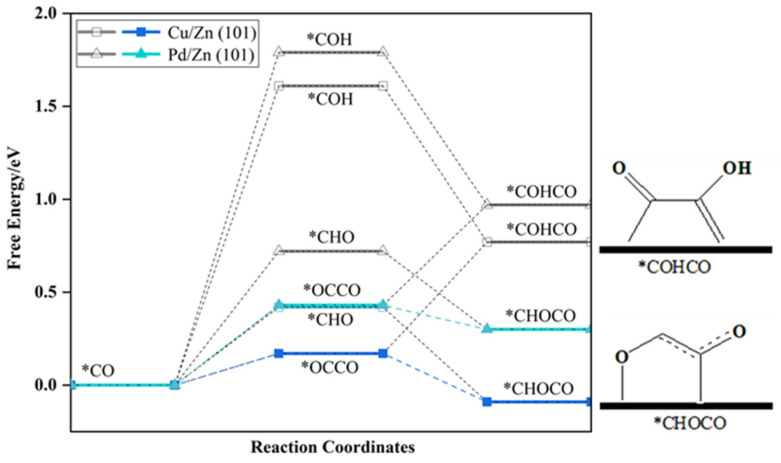
Free Energy Diagrams of possible C–C coupling mechanisms on Pd/Zn (101) (triangles) and Cu/Zn (101) surfaces (squares), where the cyan line represents the most efficient reaction mechanism on the Pd/Zn (101) surface, and the blue line represents the optimal reaction mechanism on the Cu/Zn (101) surface.

**Figure 4 molecules-28-04606-f004:**
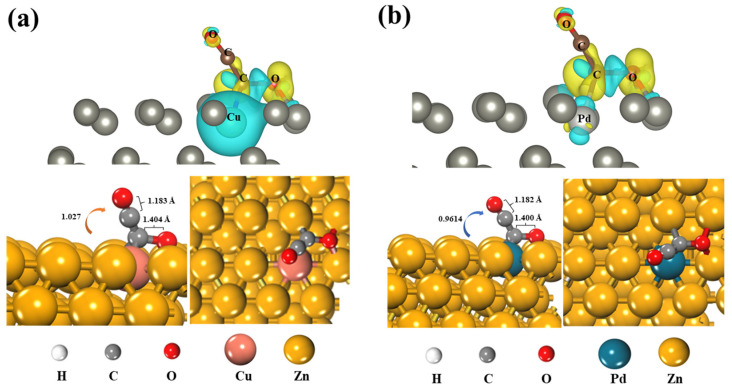
Adsorption structure and charge difference map (Iso-surface level: 0.006) of OCCO on (**a**) Cu/Zn (101) and (**b**) Pd/Zn (101), with the yellow part representing the gain of charge and the cyan part representing the loss.

**Figure 5 molecules-28-04606-f005:**
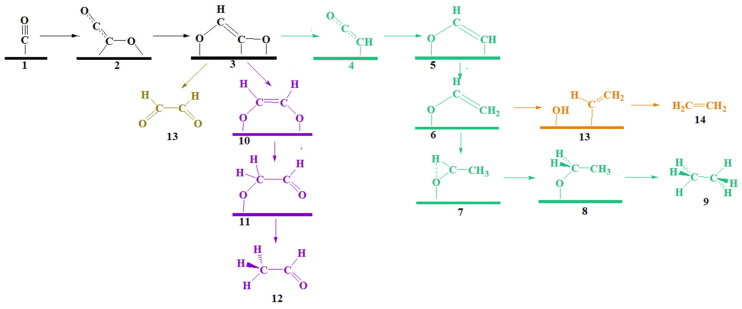
Mechanism of CO reduction to C2 products on Cu/Zn (101) surface. The black pathway represents the CO reduction reaction generating *CHOCO mechanism, the chartreuse pathway represents the acetaldehyde mechanism, the purple pathway represents the ethanol mechanism, the green pathway represents the methane mechanism, and the orange pathway represents the ethylene mechanism. 1–14 represents intermediates and products during the reaction process.

**Figure 6 molecules-28-04606-f006:**
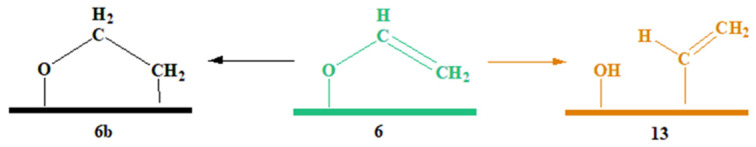
Analysis of the ethylene mechanism. The green intermediate represents the key branching intermediate for producing ethylene. Black represents the protonation path that does not occupy the thermodynamic advantage, and orange represents the thermodynamic advantage path.

## Data Availability

Not applicable.

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
