# Peer review of "CO Electroreduction Mechanism on Single-Atom Zn (101) Surfaces: Pathway to C2 Products"

_molecules, 2023, doi:10.3390/molecules28124606_

Round 1

Reviewer 1 Report

In the manuscript, Li et al. analyzed the stability and energy barriers of different reaction intermediates in the process of CO electrochemical reduction to C2 products on Zn (101) single atom alloys. The authors employ density functional theory to predict the stability, catalytic activity, and selectivity of Cu/Zn (101) and Pd/Zn (101) catalysts in an electrochemical environment at the single atom reaction site. The results provide valuable insights into the mechanism of C2 product formation and highlight the advantageous performance of the Zn (101) single atom alloy for ethane generation. After thorough evaluation, I am pleased to recommend this article for publication in Molecules following minor revisions.

1.     The introduction section should be improved, in order to provide a clear and systematic background of the current study. The author should elucidate the reason why studying the CO electroreduction mechanism instead of CO2 electroreduction.

2.     In the introduction section, the authors stated that the key to enhancing C2 production and suppressing C1/H2 production is to discover electrocatalysts that bind *CHO or *COH more strongly than *H (ΔG*H > ΔG*CHO or ΔG*COH > ΔG*OCCOH or ΔG*OCCHO). I believe that the dimerization of *CO and the formation of the *OCCO intermediate is a key step for C2 production. The authors should explain this.

3. The author addressed the stability and selectivity of Pd/Zn (101) and Cu/Zn (101) in section 2.1, but the discussion provided may be insufficient. To encourage further exploration of the theoretical findings through experimental studies, it is important to enhance the perspectives presented.

4. The authors should clarify the meaning of the arrows in Figure 3? It is recommended to use a different description method or provide a more detailed explanation.

5. In section 2.2, the sentence “The Bader results indicate that there is a total charge transfer of approximately 1 e for the four atoms directly interacting with *OCCO, and the adsorption state of the dimer has 1 e more electrons than the gas state.” is confusing, need to be rephrased.

6. It is suggested to utilize distinct colors for different atoms in Figure 4 to enhance reader comprehension.

Moderate editing of English language required

Reviewer 2 Report

Carbon dioxide reduction reaction (CO2RR) is a promising approach to address the CO2 emission problem, which has attracted much more attention from a large number of scientists working in catalysis field. This work well predicted the stability and catalytic activity of Cu/Zn (101) and Pd/Zn (101) catalysts in the electrochemical environment for CO2RR. The mechanism of C2 products (glyoxal, acetaldehyde, ethylene, and ethane) produced by electrochemical CO2RR on the surface at the single atom reaction site was well studied. In this process, the formation of *CHOCO intermediate is key step of CO2RR to generate C2 products. This work is well written and organized, which can be published in Molecules after minor revision. Some detailed comments are as follows:

1.     The *COHCO intermediate is a key step of C2 product generation as mentioned by authors. The density of states of chemical interaction between Cu/Zn (101), Pd/Zn (101) and *COHCO should be analyzed, which can provide the data why Cu/Zn (101) possessed the low adsorbed energy of *COHCO during CO2RR.

2.     Based on the series of analysis for C2 product generation over Cu/Zn (101), Pd/Zn (101), are there some key points to show the way to regulate the selectivity of CO2RR to generate some specific products? such as ethylene, ethane or others.

3.     In the model of Cu/Zn (101), Pd/Zn (101), it is easily observed that single Cu and Pd atom embedded in Zn metals, which is similar to the case of single metal-atomic electrocatalysts. Therefore, some important refs should be added in the introduction part (Nat. Energy 2019, 4, 732-745; Adv. Mater. 2021, 33, 2003075; ACS Energy Lett. 2020, 5, 1044-1053; Adv. Funct. Mater. 2020, 30, 1907658).

Carbon dioxide reduction reaction (CO2RR) is a promising approach to address the CO2 emission problem, which has attracted much more attention from a large number of scientists working in catalysis field. This work well predicted the stability and catalytic activity of Cu/Zn (101) and Pd/Zn (101) catalysts in the electrochemical environment for CO2RR. The mechanism of C2 products (glyoxal, acetaldehyde, ethylene, and ethane) produced by electrochemical CO2RR on the surface at the single atom reaction site was well studied. In this process, the formation of *CHOCO intermediate is key step of CO2RR to generate C2 products. This work is well written and organized, which can be published in Molecules after minor revision. Some detailed comments are as follows:

1.     The *COHCO intermediate is a key step of C2 product generation as mentioned by authors. The density of states of chemical interaction between Cu/Zn (101), Pd/Zn (101) and *COHCO should be analyzed, which can provide the data why Cu/Zn (101) possessed the low adsorbed energy of *COHCO during CO2RR.

2.     Based on the series of analysis for C2 product generation over Cu/Zn (101), Pd/Zn (101), are there some key points to show the way to regulate the selectivity of CO2RR to generate some specific products? such as ethylene, ethane or others.

3.     In the model of Cu/Zn (101), Pd/Zn (101), it is easily observed that single Cu and Pd atom embedded in Zn metals, which is similar to the case of single metal-atomic electrocatalysts. Therefore, some important refs should be added in the introduction part (Nat. Energy 2019, 4, 732-745; Adv. Mater. 2021, 33, 2003075; ACS Energy Lett. 2020, 5, 1044-1053; Adv. Funct. Mater. 2020, 30, 1907658).

Author Response

Although question 3 is answered in the attachment, the specific supplement is:  “In addition, SAAs have garnered significant attention in the scientific community due to their exceptional catalytic activity, which surpasses that of traditional catalysts [2]. By modifying the neighboring single atomic sites, SAAs has the ability to enhance the selectivity of CO2RR[3, 4]”. Thank you again for your helpful advice.

Reviewer 3 Report

I have completed my review of the manuscript titled "CO electroreduction mechanism on single-atom Zn (101) surfaces: pathway to C2 products" by Yixin Wang et al. Overall, I find the manuscript to be well-written and the research presented is of interest to the field of electrocatalysis for CO2 reduction. However, there are a few minor revisions that I believe should be addressed before considering the manuscript for publication. Please find my detailed comments and suggestions below:

1.     The introduction provides a good background on the importance of electrocatalytic CO2 reduction, and the challenges associated with C2 product selectivity. However, it would be beneficial to include a brief discussion on the limitations of current catalysts and the motivation for exploring single-atom alloys (SAAs) as potential alternatives. This would help contextualize the significance of the research and its contribution to the field.

2.     The authors discuss the electrochemical stability of Cu/Zn (101) and Pd/Zn (101) catalysts but do not provide a clear comparison between the two. It would be useful to include a direct comparison of the electrochemical stabilities of these catalysts, highlighting their relative advantages and potential drawbacks.

3.     The discussion on the C-C coupling mechanisms is informative, and the presented mechanisms are well-supported by the calculations. However, it would be helpful to include a brief discussion on the experimental evidence or previous studies that support these proposed mechanisms. This would strengthen the credibility of the presented mechanisms and provide a broader perspective on the existing knowledge in this area.

4.     The conclusion should be expanded to include a summary of the main findings and their implications. It would be beneficial to clearly state how the results contribute to the design of more efficient CO2 catalysts and address the global environmental and energy challenges mentioned in the introduction.

Overall, I believe that this manuscript presents valuable insights into the CO electroreduction mechanism on single-atom Zn (101) surfaces and their pathway to C2 products. With the suggested revisions, the manuscript will make a significant contribution to the field of electrocatalysis for CO2 reduction.

Minor proofreading is required.

Reviewer 4 Report

In this work, the authors performed DFT calculations to investigate the single-atom catalysis process of C2 products. By calculating the free energy, the preferred pathway and the potential mechanism is proposed. The Zn (101) single atom alloy demonstrates superior performance in generating ethane at lower voltages, while acetaldehyde and ethylene exhibit significant potential. The study's findings provide a theoretical foundation for designing more efficient and selective catalysts for carbon dioxide reduction, which can be helpful for the material science community and the green chemistry industry.

As such, the proposed article deserves to be published and the Molecules is certainly well targeted. Before the publication, I would like to ask the authors to consider the minor comments below.

1. In this work, Were all calculations done with T=298.15K? If yes, can the authors discuss if the conclusions large change at different temperature.

2. In the calculations, were the geometries at different steps optimized? Can the authors specify this in the computational details.

3. Page 3, Figure 1

Because the authors mentioned that “Our calculated results agree well with the experimental results”. I suggest add experiment reference to the Figure 1 for comparison.

4. Page 6, Figure 4

The color bar for the iso-surface is missing.

5. Page 9, line 353

“The Perdew-Burke-Ernzerhof (PBE) exchange-correlation function and projector augmented wave pseudo-potentials were empolyed in the calculations”

It is well known that the GGA functional PBE significantly underestimates the energy barrier because of the delocalization error. Can the authors discuss that will the conclusion be changed if different functionals (for example, hybrid function B3LYP) are used?

6. Page 9, line 355

“we sampled the Brillouin zone using a 3×3×1 k-points mesh”

The convergence test for the k-mesh is needed.

I noticed several typos, for example:

Page 9, line 353

“The Perdew-Burke-Ernzerhof (PBE) exchange-correlation function and projector augmented wave pseudo-potentials were empolyed in the calculations”

"empolyed" should be "employed".

The authors need to double check.
